# The Perioperative Biochemical and Clinical Considerations of Pheochromocytoma Management

**DOI:** 10.3390/ijms26136080

**Published:** 2025-06-25

**Authors:** Alexa J. Gombert, Alexandra M. Nerantzinis, Jennifer Li, Weidong Wang, Isaac Y. Yeung, Ana Costa, Sergio D. Bergese

**Affiliations:** 1Department of Anesthesiology, Renaissance School of Medicine, Stony Brook University, Stony Brook, NY 11794, USAana.costa@stonybrookmedicine.edu (A.C.); 2Renaissance School of Medicine, Stony Brook University, Stony Brook, NY 11794, USA

**Keywords:** pheochromocytoma, paraganglioma, hypertensive crisis, alpha-blockade, beta-blockade

## Abstract

Pheochromocytoma, a rare catecholamine-secreting tumor, poses significant perioperative challenges due to its potential for severe hemodynamic instability. Careful management of patients with pheochromocytoma is critical for patient safety and favorable outcomes. The diagnostic workup focuses on biochemical analysis of plasma or urinary metanephrines, followed by imaging for tumor localization and genetic testing to identify hereditary syndromes. Preoperative management emphasizes adequate alpha-adrenergic blockade followed by beta-blockade to stabilize cardiovascular function. Anesthetic planning requires meticulous attention to volume status, cardiovascular optimization, and intraoperative monitoring to mitigate the risks of hypertensive crises and hypotension. Postoperative care must account for ongoing hemodynamic and metabolic fluctuations. A multidisciplinary, protocol-driven approach is essential to improve outcomes in patients undergoing pheochromocytoma resection. This paper provides a comprehensive overview of the genetic, biochemical, clinical, and anesthetic considerations involved in the diagnosis and perioperative management of pheochromocytoma.

## 1. Introduction

Pheochromocytomas are rare neuroendocrine tumors that arise from chromaffin cells that develop within the adrenal glands, specifically in the adrenal medulla. Chromaffin cells are a specialized form of sympathetic neurons and are responsible for producing and secreting catecholamines into the bloodstream to exert physiological response to stimuli [1]. Pheochromocytomas can cause the adrenal glands to release excessive amounts of catecholamines, leading to symptoms such as high blood pressure (often severe or episodic), headaches, sweating, tachycardia, anxiety, and panic attacks [2,3]. The estimated incidence of pheochromocytomas is 0.05%, occurring primarily in adults ages 30–50, although they can occur at any age, appear with similar frequency in both adrenal glands, and are found equally in both sexes [4,5,6,7]. Pheochromocytomas are also found in 0.05% to 0.1% of patients at autopsy, indicating some cases may go completely undiagnosed [8,9,10]. Additionally, a growing number of patients with pheochromocytomas are identified incidentally during imaging for other reasons (called incidentalomas), suggesting that the prevalence of these neuroendocrine tumors may be underestimated [11,12]. Pheochromocytomas can arise sporadically or be part of a familial cancer syndrome. Roughly 30% to 40% of pheochromocytomas are associated with hereditary genetic syndromes, such as multiple endocrine neoplasia type 2 (MEN2), Von Hippel–Lindau (VHL) disease, neurofibromatosis type 1 (NF1), and paraganglioma syndromes [13,14]. Genetic counseling and family screening are now standards of care in hereditary cases, which is imperative for prognostication and treatment.

## 2. Genetic and Molecular Insights

Genetic factors play a major role in pheochromocytomas, which have the strongest link to inherited pathogenic variants (PVs) among all solid tumors [15]. Around 30–40% of cases involve germline mutations, often inherited in an autosomal dominant pattern [16]. These mutations are linked to hereditary syndromes like VHL, NF1, and MEN2 and can lead to tumors in multiple tissues, including pheochromocytomas and paragangliomas. Identification of novel genetic and molecular driver mutations has led to new insights into genomic testing, tumor pathogenesis, and personalized treatment strategies, particularly for metastatic cases. Based on specific mutations, pheochromocytoma tumors are now classified into three molecular clusters, namely (1) pseudohypoxia, (2) kinase signaling, and (3) Wingless and Int-1 Genes (*Wnt*) signaling. Understanding these clusters and their associated clinical, biochemical, and imaging signatures allows for more targeted therapies.

### 2.1. Cluster 1 (Pseudohypoxia)

Gene mutations in this cluster affect the hypoxia-inducible factor (HIF) signaling pathway. This pathway regulates the expression of genes involved in various adaptive responses to hypoxia, including angiogenesis, erythropoiesis, and metabolic adaptation. Dysfunction of the HIF-signaling pathway can result in unchecked cell proliferation [17]. The VHL gene, a tumor suppressor gene, is one of the upstream regulators in the HIF signaling pathway. Loss-of-function mutations in VHL lead to unchecked HIF activation, promoting abnormal vascular proliferation and tumor development. Germline mutations in VHL are associated with hemangioblastomas in the brain and spinal cord, renal cell carcinoma in kidney, pancreatic neuroendocrine tumors, pheochromocytoma/paragangliomas, and other tumors in different parts of the body [18]. Succinate dehydrogenase (SDH) is a mitochondrial enzyme complex involved in cellular respiration and oxygen sensing. Mutations in the SDH genes (4 subunits: SDHA, SDHB, SDHC, and SDHD) lead to increased HIF activity and stability, primarily of hypoxia-inducible factor2 alpha (HIF-2α) [19]. Tumors in this cluster often exhibit a noradrenergic biochemical phenotype and carry a higher risk of metastasis.

### 2.2. Cluster 2 (Kinase Signaling)

MEN2 is driven by constitutive activation of the Rearranged During Transfection (RET) proto-oncogene. Persistent RET expression enhances the activity of the phosphoinositide 3-kinase signaling pathway, reducing apoptosis and increasing cell proliferation [20]. Approximately 50% of MEN2 patients develop bilateral pheochromocytomas [21]. The phenotypic penetrance and age of onset of pheochromocytomas correlate strongly with specific RET codon mutations [22,23]. As a result, the American Thyroid Association recommends initiating annual lifetime screening at age 11 for individuals harboring high-risk variants such as M918T, C634, and A883F [24].

NF1 is an autosomal dominant condition associated with café-au-lait spots, neurofibromas, optic gliomas, and other tumors [25]. Although only 1–13% of NF1 patients develop pheochromocytomas, they have a high risk for the development of bilateral and metastatic pheochromocytomas [26]. NF1 is caused by null mutations of the NF1 gene, leading to hyperactivation of the mitogen-activated protein kinase signaling pathway and increased cell proliferation. Routine biochemical screening for pheochromocytomas is recommended in hypertensive NF1 patients, especially prior to surgical procedures, to mitigate the risk of hypertensive crises [27,28,29].

The transmembrane protein 127 (TMEM127) gene, located on chromosome 2q11, encodes a tumor suppressor transmembrane protein that negatively regulates the mammalian target of the rapamycin (mTOR) complex 1 pathway. The mTOR pathway is crucial for cell growth, metabolism, and protein synthesis [30]. Loss-of-function mutations in TMEM127 lead to abnormal mTOR signaling activities and tumor development, frequently linked to adrenal-based pheochromocytomas in patients, often without a family history [31]. Another pheochromocytoma-susceptible gene is MYC-associated factor X (MAX). The MAX protein interacts with the MYC complex, functioning as a transcriptional repressor [32,33]. Null mutations of MAX have been identified in familial pheochromocytoma [32,34].

Recent studies by Park et al. demonstrated that tumors classified within Cluster 2 often exhibit an adrenergic biochemical phenotype and are less likely to metastasize than Cluster 1 (12.5 vs. 38.8%; *p* < 0.01) [35]. While Cluster 1 pheochromocytomas are predominantly associated with pseudohypoxia and HIF activation, Cluster 2 pheochromocytomas can also exhibit HIF signaling. This suggests a more expansive role of HIF in the development and progression of pheochromocytomas than previously appreciated [17].

### 2.3. Cluster 3 (Wnt Signaling)

*Wnt* signaling plays a pivotal role in regulating cell differentiation and proliferation. In pheochromocytomas and paragangliomas, activation of the *Wnt*/β-catenin pathway is associated with a more aggressive tumor phenotype. The cold shock domain containing E 1 and mastermind-like transcriptional coactivator 3 genes are two somatically mutated driver genes responsible for a *Wnt*-altered subtype of pheochromocytoma. Mutations are typically observed in sporadic cases and are strongly correlated with aggressive disease and metastasis [16,36,37,38,39].

Recent advances in genomic and transcriptomic analysis have identified potential biomarkers that may influence tumorigenesis and metastatic potential. These include mutant Alpha Thalassemia/Mental Retardation Syndrome X-Linked, Trisomy 5, high Ki-67 Proliferation Index expression, mutant Solute Carrier Family 25 Member 11, Glutamic-Oxaloacetic Transaminase 2, and Dihydrolipoamide S-Succinyltransferase [40,41,42,43,44]. HIF2α-regulated Cytochrome C Oxidase Subunit 4 Isoform 2 (COX4I2) is involved in blood supply in adrenal tumors. Overexpression of COX4I2 is notable in tumors harboring VHL or SDH loss-of-function mutations, highlighting its potential as a biomarker of pheochromocytoma [45]. Micro non-coding ribonucleic acid (RNA) have also been identified as potential diagnostic and prognostic markers in pheochromocytoma. For example, microRNA-210 overexpression is seen in pseudohypoxia-related pheochromocytoma, whereas microRNA-21-3p levels correlate with mTOR activation, potentially predicting response to mTOR inhibitors [46,47,48]. Exosomal double-stranded deoxyribonucleic acid (DNA), usually detected in liquid biopsies, is also being investigated for predicting malignancy [49]. Circulating tumor DNA, released from tumor cells into the bloodstream, carries the same mutations found in the original tumor cells. Its analysis enables early diagnosis, prognosis, and real-time monitoring of treatment response through the detection of genetic change and epigenetic modification [50]. These new genetic markers, along with the susceptibility genes like RET, NF1, VHL, SDH, and fumarate hydratase, are expanding our understanding of the genetic basis of pheochromocytoma.

## 3. Biochemical Considerations

Understanding the biochemical behavior of pheochromocytomas is critical to guiding both diagnosis and perioperative management. Accurate identification through biochemical markers, genetic testing, and imaging informs risk assessment and treatment planning. Thus, preoperative biochemical control is essential to minimize the impact of catecholamine surges during surgery. Inadequate preparation can lead to serious anesthetic and surgical complications, including hypertensive crises and cardiovascular instability.

### Biochemical Markers and Diagnostic Testing

Levels of catecholamines and their metabolites should be measured if there is a suspicion of pheochromocytoma. The adrenal medulla primarily secretes the catecholamines epinephrine and norepinephrine. Epinephrine is metabolized to metanephrine via catechol-O-methyltransferase, while norepinephrine is converted to normetanephrine through the same pathway. These metabolites are further degraded by monoamine oxidase to produce vanillylmandelic acid [51]. The preferred laboratory test to identify these biochemical markers can vary by institution. In the case of a high index of suspicion, measuring plasma fractionated metanephrines is preferred as its negative predictive value is high and the test is simple [52]. In the case of a low index of suspicion, measuring 24-h urinary fractionated catecholamines and metanephrines is preferred as the plasma counterpart has a higher rate of false positives [53]. The accuracy of these tests depends significantly on the timing of the sample collection. Collecting samples at one point in time may fail to detect elevated levels as it could represent a period of low levels of secretion.

Following biochemical testing, radiologic imaging is utilized to locate the tumor with the use of computed tomography (CT) with contrast or magnetic resonance imaging (MRI) of the abdomen and pelvis in patients with biochemically proven pheochromocytomas. Additional imaging of the head, neck, and thorax should be considered in cases where there is suspicion of metastatic disease [54]. Should those modalities not succeed, total body nuclear imaging can be used to detect the tumors, such as gallium Ga-68 dotatate positron emission tomography, fludeoxyglucose-positron emission tomography, and iobenguane I-123 (also known as metaiodobenzylguanidine [MIBG]) scintigraphy [55]. When considering the superiority of one imaging modality over another, it is important to note that CT has 93–100% sensitivity in detecting intra-adrenal tumors >0.5 cm, which decreases to 90% for extra-adrenal tumors >1 cm [54]. MRI is also just as proficient at localizing pheochromocytomas. The added benefit of MRI utilization is visualization of surrounding vasculature, which can facilitate surgical planning [54]. CT scans are more commonly used in practice due to their lower cost and ease of accessibility. However, false positives on CT and MRI are not uncommon as they share a specificity in the range of 50–90% [56].

Additionally, genetic testing plays a critical role in the work-up of pheochromocytomas. About 30–40% of cases are the result of an underlying autosomal dominant familial disorder such as VHL (mutations in the *VHL* tumor suppressor gene), MEN2 (mutations in the *RET* proto-oncogene), and NF1 (mutations in the *NF1* gene) [13,14]. The frequency of pheochromocytoma is 10–20% in VHL, 50% in MEN2, and 3% in NF1, with bilateral adrenal pheochromocytomas being the most common [57,58,59,60]. Testing usually occurs after surgical resection of the tumor and confirmatory pathological diagnosis [61].

## 4. Clinical Considerations

The anesthetic management of pheochromocytomas demands a structured approach due to the tumor’s unpredictable catecholamine release and associated hemodynamic risks. Careful preoperative optimization, precise intraoperative control, and careful postoperative monitoring are essential to ensure patient safety. Advances in targeted therapies further inform perioperative planning and long-term management strategies. Figure 1 illustrates a detailed summary of the perioperative management strategies, including preoperative, intraoperative, and postoperative considerations.

### 4.1. Preoperative Assessment and Optimization

A multisystem approach is needed to ensure adequate optimization of a patient with pheochromocytoma in preparation for surgical resection. A comprehensive clinical evaluation, including a thorough patient history and physical exam, prior to surgery is necessary for ensuring optimal perioperative and postoperative outcomes. Preparation usually takes 10 to 14 days before the scheduled surgery, with the use of medical therapy to control hypertension, tachycardia, and correct intravascular volume [55]. Pharmacologic preparation is essential for all patients with catecholamine-secreting tumors and is usually selected by an endocrinologist. As there is no adopted universal standard for preparation, various approaches have been employed such as combined alpha- and beta-adrenergic blockade, calcium channel blockers, and the use of metyrosine (alpha-methyl-para-tyrosine) [62].

In 1982, Roizen et al. created a framework guideline for assessing the adequacy of preoperative alpha blockade encompassing the following: (1) No in-hospital blood pressure > 160/90 mmHg for 24-h prior to surgery (2) No orthostatic hypotension with blood pressure < 80/45 mmHg (3) No ST segment or T wave changes for one-week prior to surgery and (4) No more than five premature ventricular contractions per minute. The above criteria have consistently been shown in the literature to improve morbidity and mortality outcomes when used for preoperative assessment of adequate alpha blockade [63]. Roizen’s group also reported that the implementation of alpha blockade alone reduced mortality from 13–45% to 0–3% [63].

To initiate alpha blockade, selective alpha-blockers, such as doxazosin, are recommended. The dose is titrated until the systolic blood pressure is in the low-normal range for that specific patient’s age [64]. Phenoxybenzamine, which is a non-selective alpha-blocker, may also be used. Some studies suggest it provides superior intraoperative blood pressure control compared to selective alpha-blockers, but it carries a higher cost, with ultimately comparable overall outcomes to selective agents [64]. Once adequate alpha-adrenergic blockade is achieved, beta-adrenergic blockade is introduced approximately 2 to 3 days preoperatively. Beta-blockers should never be initiated before alpha-blockade, as unopposed alpha-receptor stimulation could lead to further elevations in blood pressure [62]. Other options for blood pressure control include calcium channel blockers, such as nicardipine and amlodipine. They are typically used to help control blood pressure when alpha- and beta-blockers are not sufficient, or if the patient reports adverse side effects [65]. Metyrosine blocks catecholamine production via inhibition of tyrosine hydroxylase and is usually utilized when the usual alpha- and beta-blocker approach must be stopped due to poor tolerance or significant side effects [66]. Table 1 compares various features of non-selective versus selective agents for alpha-blockade.

Cardiac evaluation in patients undergoing pheochromocytoma resection is critical due to the risk of catecholamine-induced cardiomyopathy, arrhythmias, and myocardial ischemia. Preoperative echocardiography is recommended to assess left ventricular function and wall motion abnormalities, as chronic catecholamine exposure can lead to dilated cardiomyopathy with varying degrees of heart failure. A preoperative echocardiogram can also reveal moderate to severe left ventricular hypertrophy alongside varying degrees of diastolic dysfunction that directly reflect the severity, duration, and degree of blood pressure control [63]. In patients with reduced ejection fraction or diastolic dysfunction, intraoperative management should prioritize preload maintenance, avoidance of myocardial depressants, and use of inotropic support as needed [67]. Electrocardiogram (ECG) monitoring may reveal nonspecific ST-T changes, QT prolongation, or arrhythmias, findings that help guide anesthetic drug selection and perioperative rhythm surveillance [68]. Ultimately, individualized cardiac risk stratification and anesthetic planning are key to minimizing perioperative cardiovascular morbidity.

Furthermore, it is important to assess cardiovascular status, adequacy of alpha and beta blockade, volume resuscitation, and any other comorbidities. Due to catecholamine release, which causes vasoconstriction, pheochromocytoma patients often present with contracted intravascular volume. To help mitigate intraoperative hemodynamic instability, adequate resuscitation of patients with fluids preoperatively is crucial. Expanding the patient’s intravascular volume helps to prevent severe or prolonged hypotension intraoperatively after tumor resection and postoperatively [69].

Intraoperative and postoperative complications can be significant if adequate perioperative planning is not attained. The most common intraoperative complications are unstable blood pressure and heart rate [55]. Patients often experience these perturbations during anesthesia induction and with any tumor manipulation, such as tumor ligation, which can lead to a hypertensive crisis. These episodes are induced by a catecholamine secretion surge and can have serious consequences, including myocardial ischemia, arrhythmias, and stroke. These complications can be anticipated, to an extent, if there is a higher preoperative plasma norepinephrine concentration and a larger tumor size (>4 cm) [70]. These implications were further reiterated by Kiernen et al., who found that in 91 patients undergoing pheochromocytoma resection, tumor size, open adrenalectomy, and type of alpha blockade used were the highest predictors of hemodynamic instability (systolic blood pressure > 200 mmHg, blood pressure greater than or less than 30% of baseline, heart rate greater than 110 beats per minute, and the need for postoperative vasopressors) [71]. Selective alpha blockade was associated with significantly more episodes of intraoperative hypertension but no perioperative adverse outcomes. Following the removal of the tumor, given the decreased secretion of catecholamines, the patient may suffer from hypotension and/or hypoglycemia [72].

### 4.2. Intraoperative Management

The anesthetic management of the pheochromocytoma patient is a carefully titrated balance between combatting hypertension associated with catecholamine release from the tumor followed by the potential hypotension patients experience after tumor ligation. Careful planning and invasive monitoring, such as an arterial line placement prior to induction of general anesthesia, are critical to maintain hemodynamic stability and avoid cardiac ischemia [73]. Establishing peripheral access with multiple large-bore intravenous catheters is preferable due to the possible need for rapid infusion of fluids and other medications. Central venous access is often necessary for the administration of vasoactive agents and intravenous fluids, especially in patients with poor left ventricular function [63]. Transesophageal echocardiography, although infrequently used, may be used in complex cases or patients with cardiac dysfunction to assess the intravascular volume and cardiac function in real time [73]. As with any routine surgical case, constant ECG monitoring is the standard of care; however, its utilization and proper placement in pheochromocytoma resection are even more important due to the toxic effects of catecholamines on the myocardium with possible resultant left ventricular strain, hypertrophy, bundle branch blocks, and ischemia [63]. Pre-induction administration of short-acting benzodiazepines, such as midazolam, is recommended to mitigate anxiety-related catecholamine release. Administering a long-acting benzodiazepine, such as lorazepam or diazepam, the night prior to surgery has also been shown to help control hypertensive crises at induction [63].

Another critical aspect of anesthetic management should be the adequate depth of anesthesia to mitigate the effects of sympathetic stimulation, which typically stem from a few common sources, such as induction of general anesthesia and tumor manipulation [74]. Stimulation and catecholamine release stemming from laryngoscopy and endotracheal intubation can be exaggerated in patients with excess circulating catecholamines [73]. Peritoneal insufflation during laparoscopic pheochromocytoma resection can also result in abrupt release of catecholamines. Administering a bolus of intravenous lidocaine prior to endotracheal intubation has been shown to aid in the reduction of sympathetic release [69]. Regarding induction agents, etomidate is commonly favored due to its hemodynamic stability, particularly in patients with compromised cardiac function [73]. Propofol is also used, though it may cause hypotension, especially in volume-depleted patients [63]. Ketamine should be avoided in these cases due to its sympathomimetic effects and its propensity to cause hypertension and tachycardia [63]. Adequate neuromuscular blockade can be achieved through many non-depolarizing agents, such as vecuronium, rocuronium, and cisatracurium, which are all equally effective in achieving neuromuscular blockade [75]. The depolarizing agent, succinylcholine, should be avoided due to its potential to cause catecholamine surges from the muscular fasciculations that it produces, leading to possible tumor compression. Pancuronium should also be avoided due to its propensity to cause vagus nerve inhibition with a subsequent pressor response, which in pheochromocytoma patients can lead to persistent tachycardia and/or hypertension [63]. In general, all histamine-releasing agents should be avoided as much as possible, as they release catecholamines from chromaffin granules [69]. Opioids such as fentanyl, remifentanil, and hydromorphone are often liberally used as adjuncts to blunt the hemodynamic response to laryngoscopy, intubation, and insufflation in laparoscopic resections [63]. Morphine should be avoided due to its histamine-releasing properties, ultimately increasing the risk of a hypertensive crisis [73].

Inhaled anesthetics have been the agents of choice for the maintenance phase of anesthesia for pheochromocytoma resection. The most critical period of anesthesia maintenance is during tumor manipulation when massive catecholamine release can precipitate sudden, severe hypertension, tachyarrhythmias, or even pulmonary edema [63]. Volatile anesthetics such as sevoflurane and isoflurane have been extensively used and are favorable due to their rapid titratability and depth control, but they must be carefully adjusted to avoid hypotension [63,73]. Desflurane, an otherwise popular inhaled anesthetic for its low blood-gas partition coefficient, should be avoided in pheochromocytoma patients as it possesses the potential to stimulate a significant sympathetic response. Halothane is contraindicated in pheochromocytoma resection due to its ability to sensitize the myocardium to catecholamines making it a potent arrhythmogenic [63].

### 4.3. Complications

Severe hypertension occurs often in surgical resection of these tumors, which can be accompanied by tachycardia especially in patients who were using beta blockers preoperatively [76,77]. Hypertensive crises can cause myocardial ischemia and stroke; therefore, its quick correction is crucial [73]. Depending upon the prominent catecholamine released by the tumor, epinephrine or norepinephrine, a hemodynamic crisis may present as severe bradycardia accompanied by hypertension or with severe tachycardia and tachyarrhythmias [63].

Blood pressure surges are typically managed with short-acting vasodilators such as sodium nitroprusside, nitroglycerine, nifedipine, nicardipine, and phentolamine [73,77]. Magnesium sulfate has also recently gained prominence in its utilization for intraoperative hypertension due to its ability to prevent calcium uptake from neuronal endings, blocking catecholamine release, making it a potent vasodilator [77]. Magnesium sulfate’s antiarrhythmic properties also augment its use as a treatment for ventricular arrhythmias seen in these patients. Intravenous labetalol is ideally avoided in acute intraoperative hypertension as it has an alpha-to-beta ratio of 1:7 and, therefore, shows a preference for beta receptors with resultant worsening hypertension. It may also cause persistent hypotension and bradycardia once the tumor is ligated [73]. Overcorrection should be avoided, as profound hypotension often follows adrenal vein ligation due to the abrupt cessation of catecholamine secretion, particularly in patients with extended preoperative alpha blockade. At this stage, aggressive volume resuscitation with crystalloids or colloids is critical, and vasopressors such as phenylephrine and norepinephrine are frequently required and must be titrated carefully. Vasopressin has been effective in maintaining blood pressure in cases of refractory hypotension as it does not exert any action on peripheral adrenergic receptors [73]. Ephedrine should be avoided due to its propensity to stimulate endogenous release of catecholamines. Atropine, a muscarinic agonist, should also be avoided in pheochromocytoma patients as it inhibits parasympathetic tone, leading to increased heart rate, which can worsen myocardial oxygen demand and increase the risk of arrythmias or cardiovascular instability.

A high degree of blood pressure variability, especially sudden surges or drops, is associated with increased morbidity, including cardiac ischemia, arrhythmias, and end-organ damage. These hemodynamic instabilities can also contribute to higher mortality rates, particularly if not promptly managed [73]. Thus, minimizing blood pressure variability is critical for reducing adverse outcomes. Hyperglycemia may also ensue intraoperatively secondary to catecholamine excess, and insulin infusion therapy should be considered as indicated [63]. Therefore, frequent glucose monitoring intraoperatively is imperative, with readily available insulin therapy typically administered as an insulin infusion.

### 4.4. Postoperative Care

Postoperatively, patients should be monitored in an intensive care setting, as delayed hypotension, hypertension, hypoglycemia, and cardiac arrhythmias can occur. While a majority of patients become normotensive after surgical resection of the pheochromocytoma, plasma catecholamine levels do not return to baseline until 7 to 10 days post-resection due to the slow release of stored catecholamines from peripheral nerves [78]. 50% of patients remain hypertensive several days after tumor resection and are typically treated when blood pressure is greater than 180/110 mmHg with esmolol or alpha-blockers [69]. 25% to 30% of patients will remain hypertensive indefinitely after resection [78]. On the other hand, hypotension is equally prominent. The hypotension that patients may experience after pheochromocytoma resection is often multifactorial, and the underlying cause may not always be obvious. Residual hypotension may result from the downregulation of alpha-adrenergic receptors, hypovolemia, or residual effects of long-acting antihypertensive medications like phenoxybenzamine. These effects are typically seen 24–48 h postoperatively [69]. The treatment of postoperative hypotension after pheochromocytoma resection should be geared towards its underlying cause and typically includes fluid resuscitation and vasoactive agents [69]. Postoperative hypoglycemia following pheochromocytoma resection results from a sudden drop in circulating catecholamines, which previously suppressed insulin secretion and promoted insulin resistance. Tumor removal leads to unopposed insulin activity, increased peripheral glucose uptake, and reduced hepatic glucose production. Additional contributing factors include depleted glycogen stores, preoperative beta-blockade, and potential adrenal insufficiency. It is recommended that blood sugar be monitored every four to six hours [73,77].

Cardiac arrhythmias following pheochromocytoma resection should be promptly evaluated with continuous ECG monitoring, electrolyte assessment (particularly potassium, magnesium, and calcium), and cardiac biomarkers to rule out ischemia. These arrhythmias may result from residual circulating catecholamines, abrupt withdrawal of sympathetic stimulation, intraoperative myocardial strain, or postoperative hypotension and fluid shifts. Beta-blockers or antiarrhythmic agents may be required depending on the arrhythmia type and hemodynamic stability [77]. Early recognition and targeted management are essential to prevent progression to more severe cardiovascular complications. A less severe, although more common, complication of anesthesia includes postoperative nausea and vomiting. Medications such as H_2_ blockers and ondansetron are safe and effective in pheochromocytoma patients. Medications with dopamine antagonist activity, such as metoclopramide, amisulpride, and droperidol should be avoided as these medications can indirectly increase sympathetic nervous system activity and exacerbate the effects of excess circulating catecholamines [6,73]. Table 2 provides a comprehensive overview of the most commonly used perioperative medications, highlighting those considered safe versus those to avoid in patients with pheochromocytomas.

### 4.5. Targeted Therapies

Although established tests like metanephrine measurements remain important, new biomarkers hold promises for improving diagnosis, predicting metastasis, and guiding personalized treatment. Although surgery remains the first line treatment for pheochromocytomas, targeted therapies offer personalized options to improve survival rates and minimize side effects. Currently, ultratrace MIBG is the only approved radionuclide therapy for metastatic pheochromocytomas in the United States [79,80]. Understanding the molecular subtype of pheochromocytoma allows for more targeted therapies. For instance, HIF inhibitors may be effective for tumors with mutations in genes like VHL and SDH, while mTOR inhibitors could be considered for Cluster 2 tumors with mutations in genes like TMEM127. SDHB-mutant (metastatic) pheochromocytomas exhibit a high risk of malignancy, and succinate accumulation drives epigenetic dysregulation.

Some germline SDHB mutations were associated with hypermethylation of the O^6^-methylguanine-DNA methyltransferase (MGMT) promoter. When pheochromocytoma patients with hypermethylated MGMT were treated with temozolomide, a significant number of patients had improved progression-free survival [81]. Poly (ADP-Ribose) Polymerase (PARP) inhibitors, such as olaparib and talazoparib, have been investigated in combination with temozolomide to treat advanced pheochromocytomas by preventing PARP from repairing damaged DNA for the tumor cells, which can lead to cell death or slowed growth. In preclinical models of SDHB-mutant pheochromocytomas/paragangliomas, olaparib has been shown to enhance the effect of temozolomide [82]. A Phase II trial has already been initiated at multiple institutes, including the National Cancer Institute and Dana–Farber Cancer Institute, to compare the progression-free survival of patients with advanced pheochromocytoma receiving temozolomide alone to that of patients receiving temozolomide plus olaparib [83]. Peptide receptor radionuclide therapy (PRRT) may be considered as a second-line treatment or palliative option for metastatic or inoperable pheochromocytomas [84]. PRRT uses radioactive materials (Lutetium-177) combined with a somatostatin analog (octreotide or octreotate) to target and destroy somatostatin receptor-positive pheochromocytomas. The radiopharmaceutical Lutetium-177 DOTA-Tyr3-Octreotate (Lu-177) dotatate is administered intravenously, allowing tumor-specific binding of radioactive-labeled somatostatin analog to its receptor. PRRT has shown good efficacy in controlling tumor growth and progression in some patients with advanced pheochromocytoma and paraganglioma [85,86,87]. A phase II trial at the National Cancer Institute is recruiting metastatic/inoperable patients to evaluate the safety and tolerability of Lu-177-dotatate to see if it improves the length of time it takes for the cancer to return [88]. Everolimus, an mTOR inhibitor, has shown some efficacy in reducing lesion size in a subset of pheochromocytomas such as NF1 patients [89]. Selpercatinib, a potent small molecule RET kinase inhibitor, has demonstrated remarkable antitumor activity in selective RET-activated solid tumors. Case reports undertaken by Barbars Deschler-Baier et al. have demonstrated that targeted RET inhibition by selpercatinib is an effective therapy against RET-mutant pheochromocytoma [90]. Belzutifan is a promising HIF-2α inhibitor being studied in clinical trials for pheochromocytomas and is currently the only HIF-2α inhibitor therapy approved in the United States for the treatment of patients with VHL-related tumors [91,92]. Other tyrosine kinase inhibitors (TKIs), such as sunitinib, axitinib, and cabozantinib may also be used as targeted therapies for metastatic or inoperable pheochromocytomas due to their antiangiogenic activities [93]. Both sunitinib and cabozantinib have been evaluated in clinical trials for progressive metastatic pheochromocytoma/paraganglioma, with some promising results suggesting they may be considered as first-line therapy [94,95]. Pazopanib hydrochloride and Lenvatinib are TKIs currently being tested in clinical trials as antineoplastic/anti-angiogenesis candidates for advanced pheochromocytoma at the Mayo Clinic (Rochester, MN and Jacksonville, FL, USA) [96,97,98].

Targeting epigenetic modulators has emerged as a promising strategy for pheochromocytoma therapy. DNA methylation, a process where methyl groups are added to DNA, can regulate target gene expression. Mutations in genes involved in epigenetic regulation, such as those encoding DNA methyltransferases and histone modifiers, have been reported in pheochromocytoma cases [99,100]. Differential methylation patterns in the promoters of genes encoding catecholamine synthesis enzymes result in diverse phenotypes of pheochromocytomas due to the differential ratio of metanephrine/normetanephrine secretion [101]. Vorinostat, a histone deacetylase (HDAC) inhibitor, has shown promise in pheochromocytoma imaging and therapy for potentially increasing the uptake of radioactive tracers like 123I-MIBG [102]. Other HDAC inhibitors like belinostat and romidepsin are also being explored as potential therapeutic agents for various cancers, including advanced pheochromocytoma [103,104].

Immunotherapy harnesses the body’s own immune system to fight cancer cells, which offers a new approach to treating pheochromocytomas, particularly for metastatic or advanced cases where traditional therapies like surgery and chemotherapy have limited effectiveness. Currently, different immunotherapy strategies, including intratumoral immunotherapy, checkpoint inhibitors, and Chimeric Antigen Receptor T-Cell Therapy (Car-T), are being explored and show promising results in preclinical and clinical trials. Intratumoral immunotherapy involves injecting a combination of immune-stimulating agents directly into the tumor to trigger a robust response that can eradicate tumor cells [105]. For instance, Mannan-BAM, Toll-Like Receptor (TLR) Ligands, and Anti-Cluster of Differentiation 40 (CD40) Antibody Therapy (MBTA) Therapy work by injecting mixed components of Mannan-BAM, TLR-agonists and Anti-CD40 antibody into cancerous tissues. TLR agonists such as lipoteichoic acid or resiquimod aim to boost the immune system, while Anti-CD40 antibody is utilized to stimulate B-cells for enhancing/maintaining the immune response. Mannan–Bam is a component derived from yeast cell walls. Its pathogen-associated molecular pattern helps to present TLR-agonists and anti-CD40 antibodies to the tumor (a drug delivery vehicle) [106]. While MBTA therapy is still in the preclinical stage, its effectiveness in animal models of human pheochromocytoma raised the possibility of its use for treating advanced or metastatic cases. A study performed by Uher et al. [104] reported that MBTA can successfully recruit both innate and adaptive immune cells to the tumor tissue. Prolonged immune responses lead to a complete eradiation of tumors in a significant number of pheochromocytoma mice [107]. Checkpoint inhibitors such as nivolumab and pembrolizumab work by blocking Programmed Cell Death Protein 1/Programmed Death Ligand 1 proteins that can help cancer cells evade the immune system [108]. Clinical trials utilizing nivolumab and pembrolizumab with these checkpoint inhibitors in pheochromocytomas have shown limited but encouraging results, with some patients demonstrating response to drugs and disease stabilization [109].

Researchers are also exploring other strategies by combining checkpoint inhibitors with other therapies such as chemotherapy, radiotherapy, angiogenesis inhibition, or tumor antigen-specific immunotherapy. Car-T cell therapy, a form of immunotherapy, involves modifying a patient’s T cells to recognize and attack cancer cells. Clinical studies have been initiated using the Car-T technique to target tumor-associated antigens like Interleukin 13 Receptor Alpha 2 and Carbonic Anhydrase IX [110,111]. In sum, ongoing studies are focused on understanding the molecular and cellular mechanisms of immune involvement in pheochromocytoma, optimizing immunotherapy approaches, and tailoring strategies to improve response rates and reduce toxicity. Combining different targeted therapies, such as TKIs with immunotherapy or PARP inhibitors, may help clinicians to achieve better outcomes.

## 5. Conclusions

The perioperative management of pheochromocytomas presents unique challenges due to the tumor’s catecholamine-secreting nature and the potential for life-threatening hemodynamic instability. During pheochromocytoma resection surgery, the anesthetic management requires continuous anticipation and rapid response to extreme and sudden hemodynamic fluctuations. A thorough understanding of the biochemical profile of the tumor, coupled with vigilant preoperative optimization, is essential to minimize perioperative risks. Preoperative alpha-adrenergic blockade, preferably with agents such as phenoxybenzamine or doxazosin, remains the cornerstone of preparation, reducing perioperative morbidity and mortality by minimizing hypertensive crises and improving intravascular volume status. Beta-blockade may be cautiously introduced only after adequate alpha blockade to prevent unopposed vasoconstriction. Additionally, biochemical markers, genetic testing, and imaging can guide risk assessment and treatment planning.

Intraoperatively, continuous hemodynamic monitoring, adequate central and peripheral intravenous lines, and rapid access to vasoactive medications are essential due to the risk of severe fluctuations in blood pressure during tumor manipulation. Postoperative care involves close observation for hypotension, hypoglycemia, and arrhythmias, which can occur as catecholamine levels rapidly decline. Despite progress in the perioperative management of pheochromocytoma, significant gaps persist, particularly in the absence of high-quality randomized controlled trials to inform standardized protocols. For example, the optimal duration and titration strategy of preoperative alpha-adrenergic blockade remains unclear, with current practices based largely on retrospective data and expert opinion. Special populations—such as pediatric patients, individuals with metastatic disease, or those with hereditary syndromes like MEN2—are frequently excluded from studies, resulting in a lack of stratified guidelines tailored to their unique clinical needs. Additionally, while genetic and molecular markers offer the potential for risk stratification, they have yet to be meaningfully incorporated into perioperative planning. Targeted therapies offer promising advances in the management and treatment of pheochromocytomas, leading to improved survival rates. As surgical techniques and anesthetic strategies advance, a collaborative approach amongst anesthesiologists, endocrinologists, surgeons, and genetic counselors remains key to optimizing outcomes for these high-risk patients.

## Figures and Tables

**Figure 1 ijms-26-06080-f001:**
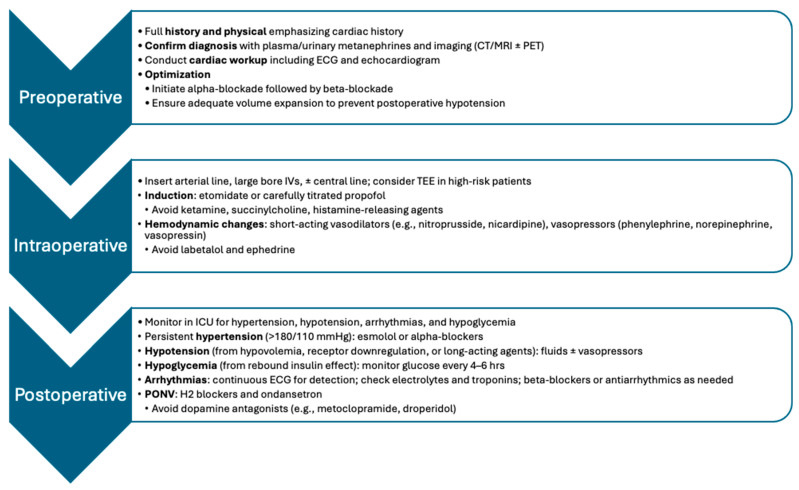
Summary of the perioperative management of pheochromocytoma patients. © [Alexa Gombert] via Microsoft Word SmartArt.

**Table 1 ijms-26-06080-t001:** Comparison of preoperative non-selective vs. selective alpha blockade.

	Non-Selective Alpha-Blockers	Selective Alpha-Blockers
Drug	Phenoxybenzamine	Doxazosin Prazosin
Mechanism	Irreversible blockade of α_1_ and α_2_ receptors	Reversible blockade of α_1_ only
Pharmacodynamics	Slower onset Longer half-life	Faster onset Shorter half-life
Hemodynamics	Reflex tachycardia (due to α_2_ blockade) Postoperative hypotension (due to longer half-life) Better intraoperative blood pressure control	Less reflex tachycardia and postoperative hypotension
Accessibility	More expensive Lower availability	Less expensive Greater availability
Indications	Large, secreting tumors Longer optimization	Shorter surgeries Faster optimization

**Table 2 ijms-26-06080-t002:** Summary of medication safety in pheochromocytoma patients.

Drug Class	Safe	Avoid
Antiemetics	H_2_ Blockers, Ondansetron	Metoclopramide, Droperidol, Amisulpride
Induction Agents	Propofol, Etomidate, Fentanyl	Ketamine
Neuromuscular Blockers	Vecuronium, Cisatracurium, Rocuronium	Succinylcholine, Pancuronium
Inhaled Agents	Enflurane, Isoflurane, Sevoflurane, Nitrous Oxide	Desflurane, Halothane
Anti-hypertensives and Beta-Blockers	Dihydropyridine Ca^2+^ Channel Blockers (Nicardipine, Clevidipine), Sodium Nitroprusside, Nitroglycerine, Diltiazem, Magnesium Sulfate, Phentolamine, Esmolol	Labetalol, Propranolol, Sotalol
Vasoactive Agents	Norepinephrine, Vasopressin	Ephedrine
Opioids	Hydromorphone, Fentanyl, Remifentanil	Morphine

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
