# Peer review of "The Perioperative Biochemical and Clinical Considerations of Pheochromocytoma Management"

_ijms, 2025, doi:10.3390/ijms26136080_

Round 1

Reviewer 1 Report

Comments and Suggestions for Authors

This review presents the perioperative management of pheochromocytoma. Several revisions are recommended to improve the clarity, focus, and scholarly impact of the manuscript.

  1. Multiple sections, particularly those related to alpha-blockade, intraoperative management, and postoperative care, contain overlapping information. Consolidating these elements will help streamline the manuscript and enhance readability.
  2. The authors reviewed different therapeutic strategies and pharmacologic agents, the inclusion of comparative tables (e.g., selective vs. non-selective alpha-blockers, anesthetic drugs, postoperative complications) would improve clinical usability.
  3. The section on targeted and emerging therapies is promising but underdeveloped. Please expand this section by including more clinical trial data.
  1. The postoperative care section lacks depth. Consider elaborating on the first 24–48 hours post-surgery, including hypotension mechanisms, glucose monitoring, vasopressor choice, and ICU management pathways.
  1. The manuscript requires moderate English language editing. Many sentences are overly long, particularly in the genetics section. Please simplify sentence structure for better comprehension.

6.Define acronyms such as “TME” at first mention. Use consistent terminology for drug names (e.g., “VEH” vs “vehicle”). Use scientific phrasing such as “within 24 h” instead of “in as little as 24h”. Verify that all abbreviations in figure legends are clearly defined and consistently used throughout the manuscript.

Comments on the Quality of English Language

The manuscript requires moderate English language editing. Many sentences are overly long, particularly in the genetics section. Please simplify sentence structure for better comprehension.

Reviewer 2 Report

Comments and Suggestions for Authors

This is a well-written and comprehensive review article entitled "The Perioperative Biochemical and Clinical Considerations of Pheochromocytoma Management." The manuscript offers a thorough and multidisciplinary overview of the current understanding of pheochromocytoma, including genetic subtypes, biochemical diagnosis, perioperative anesthetic strategies, and emerging targeted therapies. It is both informative and timely for the readership of IJMS.

The following points should be addressed to improve the clarity, consistency of the paper:

1. The abbreviation "SDHx" (page 4, line 144) should be defined at its first mention (e.g., "succinate dehydrogenase complex (SDHx)") to ensure clarity for readers unfamiliar with the term.

2. The sentence "In the case of a low index of suspicion, measuring 24-hr urinary fractionated catecholamines and metanephrines is preferred as the plasma counterpart has a higher rate of false positives" (page 4, lines 162-166) is grammatically correct but uses "24-hr" inconsistently with other time formats in the text (e.g., "24 hours" in page 6, line 254).

3. mixed use of “pre-operative” vs. “preoperative” (and similarly “post-operative” vs. “postoperative”). Please make them consistent.

4. HIF, PCC, PPGL, and MBTA appear before they are defined. Suggested Change: at first mention, include the full term and abbreviation, e.g.:Hypoxia-Inducible Factor (HIF); Pheochromocytoma and Paraganglioma (PCC/PGL); Mannan-BAM, TLR ligands, Anti-CD40 antibody Therapy (MBTA)
